# 3D Hepatic Organoid-Based Advancements in LIVER Tissue Engineering

**DOI:** 10.3390/bioengineering8110185

**Published:** 2021-11-14

**Authors:** Amit Panwar, Prativa Das, Lay Poh Tan

**Affiliations:** 1School of Materials Science & Engineering, Nanyang Technological University, Singapore 639798, Singapore; amit005@e.ntu.edu.sg; 2Faculty of Biotechnology, Institute of Bio-Sciences and Technology, Shri Ramswaroop Memorial University, Lucknow-Deva Road Barabanki, Uttar Pradesh 225003, India; 3The Henry Samueli School of Engineering, University of California, Irvine, CA 92617, USA; prativad@uci.edu; 4Singapore Centre for 3D Printing (SC3DP), Singapore 639798, Singapore

**Keywords:** liver tissue engineering, hepatic organoids, hepatic function, hepatocytes, 3D culture

## Abstract

Liver-associated diseases and tissue engineering approaches based on in vitro culture of functional Primary human hepatocytes (PHH) had been restricted by the rapid de-differentiation in 2D culture conditions which restricted their usability. It was proven that cells growing in 3D format can better mimic the in vivo microenvironment, and thus help in maintaining metabolic activity, phenotypic properties, and longevity of the in vitro cultures. Again, the culture method and type of cell population are also recognized as important parameters for functional maintenance of primary hepatocytes. Hepatic organoids formed by self-assembly of hepatic cells are microtissues, and were able to show long-term in vitro maintenance of hepato-specific characteristics. Thus, hepatic organoids were recognized as an effective tool for screening potential cures and modeling liver diseases effectively. The current review summarizes the importance of 3D hepatic organoid culture over other conventional 2D and 3D culture models and its applicability in Liver tissue engineering.

## 1. Introduction

The liver, accounting for 2–5% of the total body weight, is the largest internal organ present in the human body. It is responsible for various essential physiological functions including drug detoxification, synthesis of essential proteins, bile production, and regulation of the majority of the chemicals present in the bloodstream. Hepatocytes (HEPs) are the major parenchymal cells present in the liver and constitute 70% of the total liver cell population whereas hepatic stellate cells (HSCs), Kupffer cells, liver sinusoidal endothelial cells constitute the major non-parenchymal cell population. Damage to these cells by various means including drug toxicity, alcohol abuse, viral infection (Hepatitis B, Hepatitis C), or physical injury occasionally led to hepatic fibrosis and if remain untreated may lead to cirrhosis which is responsible for 1 million death worldwide per year [1]. Therefore, effective models for evaluating disease mechanisms to sketch therapeutic approaches and screen potential drugs for liver diseases are an area of prime interest. Over the years in vivo animal models have played a crucial role in understanding liver pathophysiology and preclinical evaluation of drug effectivity [2]. However, challenges in replication of organ-level physical and chemical microenvironment in two-dimensional in vitro cultures and genotypically different, ethically constraint in vivo rodent models, warrants a need to fabricate in vivo mimicking three-dimensional (3D) in vitro models while addressing healthcare solutions in the current time. Furthermore, the cells used for establishing in vitro models are primary hepatocytes which become non-proliferative, show rapidly declining cell polarity and liver-specific functions in 2D in vitro models [3]. The other research direction is the development of hepatic organoids, either derived from adult primary cells or iPSCs and also termed as microtissues that self organize in vitro by replicating in vivo mimicking mechanisms, which can preserve many hepatic functions, and have thus found applicability in both academic and clinical research [4]. The present review will focus on current progress in hepatic organoid-based liver tissue engineering for application in in vivo mimicking modeling of liver diseases, screening potential drug libraries, and other recent developments in tissue engineering approaches to achieve the goals of 3R, i.e., Replace, Reduce, and Refine.

## 2. Physiological Microenvironment of the Liver

### 2.1. Liver Physiological Units

#### 2.1.1. Liver Lobule

The hepatic lobule is an anatomical unit of the liver organized into polygons of irregular shapes and is composed of a hepatocytes plate arranged outward from the central vein to the portal triad tract. Hepatic acinus is a functional unit of the liver and has three zones with respect to the distance from the arterial blood. In zone 1, HEPs are in closest proximity with the arterioles and receive the maximum oxygen content. Zone 1 HEPs are also the first ones to get exposed to toxic substances carried by the portal vein from the gastrointestinal tract. Zone 3 HEPs are in the peripheral region of the acinus near the central vein, whereas zone 2 HEPs are interspersed between both the zones. Arterial and portal venous blood flows towards the central vein and the bile flows away from the central vein. In total, the HEPs get oxygen and nutrition from the arterial and portal vein blood for bile production which is further carried to bile canaliculi and other ducts.

#### 2.1.2. Liver Sinusoid (LSs)

Liver sinusoids are 10–15 µm cavities with physiological microenvironments localized among liver plates as shown in Figure 1. LSs demonstrate strong permeability for the material exchange between liver cells and bloodstream due to the presence of fenestrated liver sinusoidal endothelial cells. The sinusoid constitutes HEPs and liver sinusoid endothelial cells (LSECs) on opposite ends with HSCs present between them in the gap called ‘space of disse’. Kupffer cells (KCs) are fixed macrophages and are present in the vascular lumen, whereas HEPs form the bile canaliculi by fusing with one another. Altogether, the packaging of cells forming liver sinusoids is in such a way that it seems similar to a layered deposition of the cells. LSs are positioned radially in the liver lobule, which converges to form the central vein at the center of the lobule. In addition, the branches of hepatic arteries called arterioles branch into sinusoids to facilitate arterial blood to the cells.

### 2.2. Cell Types and Composition

The liver encompasses different types of primary cells to perform specialized functions and communicate through paracrine and autocrine signaling. Hepatocytes (HEPs), hepatic stellate cells (HSCs), liver sinusoid endothelial cells (LSECs), and Kupffer cells (KCs) are the primary cells that perform the essential functions of the liver.

#### 2.2.1. Parenchymal Cells/Hepatocytes (HEPs)

HEPs are large epithelial cells (20–30 µm) and are of extreme significance owing to their central functioning in the liver [6]. The major functions of HEPs are metabolic, secretory, and endocrine functions. 80% of liver mass and 60% of total liver cells are contributed by the HEPs [7]. HEPs are polygonal in shape and are polarized. Inside the liver, they are located either facing sinusoids (Sinusoidal face/sinusoidal domain) or facing neighboring HEPs (Lateral face/canalicular domain) as shown in Figure 1. Some canalicular domain HEPs are modified to form the bile canaliculi and are exposed to highly concentrated bile salts for bile formation, whereas sinusoidal HEPs have microvilli on the sinusoidal face for better absorption from the plasma [8]. The canalicular domain of HEPs is rich in ATP-binding cassette (ABC) transporters for detoxification and bile acid efflux transporters. ABC transporter deficiency would lead to cholestasis (bile secretion failure). Sinusoidal domain HEPs have different tyrosine kinase receptors such as epidermal growth factor receptor (EGFR), lipid/Iron scavenging receptors such as low-density lipoprotein receptor (LDLR), and transferrin receptor (TfR), in addition to numerous bile acid uptake transporters [9]. HEPs are enriched with organelles dedicated for metabolism (Mitochondria, peroxisomes, Golgi complexes, and endoplasmic reticulum (Rough & smooth)) and are either mononucleated or binucleated. 50 different types of cytochrome P450 enzymes are present in endoplasmic reticulum (ER) membrane of HEPs to metabolize fat-soluble toxins. Drug metabolism studies have classified metabolism into two phases—Phase 1 and Phase 2. Phase 1 involves subtype CYP 1A2, 2A6, 2C9, 2C19, 2D6, 2E1, and 3A4 of cytochrome P450 enzymes to perform oxidation, reduction, hydrolysis, and dehydrogenation of drugs, whereas Phase 2 detoxification is carried out by terminal transferases which permits sulfation or glucuronidation followed by excretion by kidneys [6]. As HEPs are central to the majority of liver functions, they have been employed for hepatic organoid formation through various techniques. For tissue engineering, HEPs have been procured either directly from the liver, called primary human hepatocytes (PHH), or are differentiated from stem cells (Embryonic stem cells (ESC)). In addition to this, hepatocarcinoma cell lines, genetically engineered cell lines, and ‘humanized’ metabolically competent hepatocytes have also been explored for their potential applications in drug testing, toxicological analysis, and disease models.

#### 2.2.2. Hepatic Stellate Cells (HSCs)

HSCs are spindle-shaped cells with an oval or elongated nucleus. They are present in the subendothelial space between hepatic plates and the anti-luminal face of endothelial cells as shown in Figure 1. They include one-third of the total non-parenchymal cells and 5–8% of total cells present in the liver. The major function of HSCs is the storage and transport of retinoids (Vitamin A compounds) [10,11]. Retinoid droplets are stored in the cytoplasm as retinyl esters or retinyl palmitate. These droplets also contain a significant number of phospholipids, cholesterol, free fatty acids, and triglycerides. Apart from this, HSCs secrete apolipoproteins and lipids such as prostaglandins. Prostaglandins play a key role in hepatic metabolism, neural-mediated vasoregulation, and inflammation [12]. It also secretes cytokines which help in signal transduction for cell–cell interaction in normal or injured liver. Transforming growth factor (TGF-α) and epidermal growth factor (EGF) secreted by HSCs are important epithelial growth factors which induce hepatocyte proliferation during liver regeneration [13]. HSCs play a vital role in liver diseases as shown in Figure 2 [14]. In alcoholic liver disease, they produce neutrophil chemoattractant which would end in neutrophil accumulation and produce complement C4 which assists in the liver’s inflammatory response [15]. They also function as antigen-presenting cells which can stimulate lymphocyte proliferation or apoptosis. During liver fibrosis, HSCs get activated by alcohol intake and viral infection. Activation leads to proliferation, migration of HSC, synthesis of collagen I and α-smooth muscle actin which would result in extensive ECM deposition. They also enhance the inflammatory response by activating mono/polymorphonuclear leukocyte infiltration. In activated state, HSCs produce chemokines (MCP)-1, CCL21, RANTES, and CCR5 [16,17]. HSCs also express toll-like receptors (TLRs) which reflect their affinity to interact with bacterial lipopolysaccharide (LPS) [18]. As they have a crucial role in liver diseases and host pathogen interactions, they have been integrated in hepatic organoids with multiple cell types for their application in disease models and toxicological studies.

#### 2.2.3. Hepatic/Liver Sinusoidal Endothelial Cells (LSECs)

LSECs are elongated endothelial cells and are present at the interface of blood cells, hepatic stellate cells, and hepatocytes. Lack of diaphragm and basement membrane, along with the association of ‘fenestrae’ makes them the most permeable cells in the mammalian body, as shown in Figure 1. They are highly specialized cells and represent a major fraction of non-parenchymal cells (48%). They possess the highest potential for endocytosis because of their continuous exposure to microbes and food-borne antigens present in sinusoidal space [20]. Apart from this, LSECs help in maintaining the non-activated state of hepatic stellate cells and prevent intrahepatic vasoconstriction as well as fibrosis development. As they express endothelial nitric oxide synthase (eNOS) protein, sensitive to blood flow rate and Vascular endothelial growth factor (VEGF), this assists in maintaining low portal pressure for the regulation of liver vascular tone, despite changes in hepatic blood flow [21]. LSECs can easily be characterized by the presence of LYVE1, STAB2, PECAM1 and CD32B markers [22]. They express TLRs which help in the detection of foreign particles and self-apoptotic products which initiate inflammatory reactions. They play an important role in the beginning and advancement of chronic liver diseases [23]. They enhance angiogenesis, vasoconstriction, undergo capillarization, and lose their preventive properties [24]. In hepatoma development and its progression, as well as liver lesions, LSECs interact with platelets and inflammatory cells [25]. As LSECs are sensitive to changes in shear stress, they get renewed from LSECs and LSECs progenitor cells after hepatectomy or acute liver injury during liver regeneration.

#### 2.2.4. Kupffer Cells (KCs)

KCs are the major element of the mononuclear phagocytic system and contribute fairly to both systemic and hepatic immune response. KCs are irregularly shaped cells from 10–13 µm size and account for 15% of the total liver cells [26]. They are present in the cavity of liver sinusoids and are attached to the LSECs surface. Therefore, they are in close contact with parenchymal as well as non-parenchymal cells of the liver to control the hepatic function in normal as well as diseased states of the liver. As a part of innate immunity their localization favors them to eradicate pathogens via phagocytosis from the blood circulation (portal/arterial). They act as the first line of defense for the pathogens/particulate/immune-active materials present in the gastrointestinal region. KCs possess enormous plasticity and can transform into a range of polarized phenotypes based upon the local environment such as M1 phenotype (proinflammatory), different M2 phenotypes (alternative) [27]. However, the KCs phenotype is regulated with the advancement in liver diseases including Non-Alcoholic Steatohepatitis (NASH), fibrosis, hepatocellular carcinoma, and alcoholic liver disease (ALD). They are responsible for the release of a variety of cytokines such as Tumor Necrosis factor (TNF-α), Interleukin-1 (IL-1), and Interleukin-6 (IL-6). In addition to this, they are also involved in antigen presentation [28].

#### 2.2.5. Biliary Epithelial Cells/Cholangiocytes (BECs)

BECs line the 3D network of channels in the liver forming the biliary tract in the liver and comprising 3–5% of the liver cells. They perform an important part in the formation of bile components and their transport to the duodenum [29]. Bile acid is secreted and transported by hepatocytes and another organic solute to the smallest ciliary radicle/cholangioles through bile canaliculi. Further, the bile gets alkalinized after passing through the BECs lined canals and is diluted by the secretory and absorption processes of BECs. BECs secretions contribute 40% of the total bile volume. In addition, they have been observed to possess the proliferative potential and a fraction of the population have plasticity due to variation in phenotypic characteristics called reparative/reactive phenotype [30]. Transition in the phenotype of BECs takes place during the diseased state [31].

## 3. Development of Hepatic Tissues: Strategies and Challenges

### 3.1. 2D Monolayer Culture

Classical 2D monolayer culture was widely practiced for in vitro modeling of adherent cells and explored for basic characterization of liver-specific functions i.e., glucose metabolism, phase I/ phase II enzyme activities, ammonia detoxification, etc. [32]. However, on the hard tissue culture plate (TCP), hepatocytes are shown to transform from growth-arrested, differentiated stage to de-differentiated proliferative stage within 48–72 h [7]. To overcome these challenges, culture plates coated with extracellular (ECM) proteins such as collagen-I/III/IV, fibronectin, laminin, or biodegradable polymers such as gelatin are applied to increase the viability of functional hepatocytes [33]. Lin et al. previously reported that rat primary hepatocyte cultured on porcine liver-derived biomatrix (LBM) coated culture plates acquire a more rounded shape (surface area: 517 ± 35.5 mm^2^) compared to the epical shape as observed in double-layer collagen-I sandwich culture (surface area: 1111.3 ± 77.3 mm^2^) and elongated de-differentiated shape on collagen-I absorbed culture dish (surface area: 3147.2 ± 214.8 mm^2^) over 35 days of study. However, a steady decrease in albumin production from 17 mg/day to 7 mg/day on LBM was observed, whereas it declines to no albumin production for cultures maintained on collagen-I absorbed polystyrene plates [34]. Again, it is widely reported that substrate stiffness influences the phenotype of the primary hepatocytes under in vitro cultures. Natarajan et al. reported a Polydimethylsiloxane (PDMS) based mechanically tunable culture platform, coated with collagen-I to maintain primary rat hepatocytes which showed 2.7-fold higher CYP activity after 7 days of culture when compared to the culture maintained on collagen-I coated stiffer tissue culture plates. Albumin, urea secretion, and quantification of cell-viability, which remained the basic parameters for hepatocyte functionality, were significantly higher for the soft PDMS substrate compared to the hard TCP cultures [35]. Overall, it was observed that hepatocytes, if cultured on substrates with stiffness around, 2 kPa, mechanically compliant to the stiffness of the healthy liver, showed better maintenance of liver-specific functions compared to the stiff surfaces which represent the stiffness of fibrotic liver (31.1 kPa) [36]. Apart from PHH, hepatocytes derived from induced pluripotent derived stem cells (iPSCs) have been used for monolayer culture. Takanori et al. have reported the formation of a vascularized liver bud after co-culturing of human umbilical vein endothelial cells (HUVECs) and human mesenchymal stem cells (hMSCs) with IPSCs derived hepatocytes on collagen IV-coated TCP. The given co-culture system further developed into a 3D liver bud after 4–6 days of co-culture and was further transplanted in a pre-formed cranial window of the mouse [37].

### 3.2. Sandwich Culture

Over the years, primary hepatocytes remained the gold standard cells used for in vitro modeling of liver pathophysiology. However, the rapid loss of liver-specific functions of non-proliferative primary hepatocytes in 2D in vitro culture limits its application for tissue engineering approaches. In 1989 Dunn et al. first reported the culture of rat primary hepatocytes between two layers of collagen gel, also known as the sandwich culture for 42 days, and showed that it better mimics the in vivo liver microenvironment compared to the monolayer culture on collagen gel [38,39]. Hepatocytes in sandwich culture can maintain in vivo mimicking liver-specific functions, such as albumin secretion, urea secretion, CYP activity, and are recognized as a gold standard to culture primary hepatocytes in vitro. It also shows the ability to form bile canalicular networks and can induce the secretion of multidrug resistance proteins. Due to these exceptional properties, Treijtel et al. applied primary rat hepatocytes in rat tail collagen-I-based sandwich culture to study intrinsic clearance of tolbutamide and established a mathematical model in relation to drug clearance per unit volume of the sandwich culture. The established mathematical model fits well with the experimental observations for the clearance of tolbutamide per unit of volume of hepatocytes [40]. Another group Zeigerer et al. reported the application of mice primary hepatocytes in collagen sandwich culture and compared it with collagen-coated monolayer culture for in vitro modeling of endocytosis which is highly regulated by hepatic polarity. Collagen sandwich culture is well established for the maintenance of hepatic polarity associated with AMP-activated protein kinase (AMPK) upon LKB1 mediated activation. The work successfully showed the repolarization of the hepatocytes after overlaying with the collagen-I gel and formation of continuous bile canaliculi network after 20–35 h of incubation (Figure 3) in collagen sandwich culture and compared with the liver sections [41]. However, cells in sandwich culture become cholestatic due to the ability of bile acid production by the primary hepatocytes, thus leading to rupture of the junction proteins [42]. Again due to downregulation of the bile salt export pump (BSEP) and the resulting change in bile acid metabolism, the system undergoes cholestasis which limits its maintenance only for 1–2 weeks [43]. In addition to collagen, polyethylene glycol (PEG) hydrogels have been explored to fabricate diseased models from co-culture of hepatocytes and non-parenchymal cells (NPCs) derived from induced pluripotent stem cells (hiPSCs). Manoj et al. have developed steatohepatitis and fibrosis model through co-culture of differentiating hESCs and iPSCs encapsulated in PEG hydrogel matrix enriched with various ligand peptides as well as matrix metalloproteinases (MMPs). Oleic acid and TGF-β were used to induce the fibrotic and inflammatory phenotype of cells. The given 3D disease model has sustained for a month and has shown peculiar characteristics of the disease model [44].

### 3.3. Scaffold Free Spheroid/Organoid Development

To overcome the challenges associated with rapid cell de-differentiation in 2D monolayer culture and cholestatic prone collagen sandwich culture, spheroidal aggregate culture of hepatocytes or multi-liver cell-associated aggregates, also known as organoid culture, were investigated for long-term maintenance of functional hepatic cells in vitro. Many approaches have been reported for scaffold-free aggregation of liver cells to spheroid or organoid culture, e.g., hanging-drop culture, micro-molding technique, spinner culture, polymeric nanospheres, non-adhesive surface approach, rocked technique, etc. [45]. Catherine et al. developed an easily scalable human primary hepatocytes (PHHs) derived spheroid culture or organoid culture in combination with non-parenchymal cells (NPCs: stellate, Kupffer, and biliary cells) at a ratio of 2:1 PHH to NPC by applying ultra-low attachment plates. The culture was maintained for 35 days. The culture showed no decrease in viability with a constant Adenosine Triphosphate (ATP) level over the 5 weeks of in vitro culture. The number of NPCs was found to remain constant in relation to PHHs and showed distant markers i.e., CD68 for Kupffer cells, Vimentin for Stellate cells, and CK19 representing the presence of biliary cells. The spheroid culture models were efficiently able to replicate the disease conditions for steatosis, cholestasis, and viral hepatitis where PHHs were infected with recombinant adenovirus prior to spheroid formation [46]. Hendriks et al. applied a similar approach to establish two different spheroid culture models, either derived from primary human hepatocytes or HepaRG cells, to assess drug-induced cholestasis (DIC) in vitro [47]. The mechanisms of chlorpromazine-induced cholestasis were studied effectively on both spheroid models, thus suggesting its effectiveness in the modeling of human liver diseases. In another work, Kobayashi et al. reported that coating of culture dishes with poly (N-p-vinylbenzyl-4-O-β-D-galactopyranosyl-D-gluconamide (PVLA) at higher lactose density (100 μg/mL) can induce round morphology of hepatocytes due to the interaction of galactose moieties with asialoglycoprotein receptors (ASGPR) present on hepatocytes surface [48,49]. Further, Tobe et al. reported epidermal growth factor (EGF) stimulated aggregation of spherical hepatocytes on PVLA substrate and the indispensable role of calcium ion in the aggregation of hepatocytes towards the formation of stable spheroids [50]. However, both non-adherent surface approach and receptor-induced aggregation of primary hepatocytes were unable to provide a uniform size of the hepatocyte spheroid/organoids, thus unable to establish a model which could be used consistently to validate the disease progression or drug metabolism effects at constant doses. In this aspect, the hanging drop model, where a constant number of cells can be loaded to the wells of defined size to obtain a specific diameter of the spheroid is becoming a more reliable approach. In 2017, Shri et al. reported the application of hanging drop technology to prepare spheroid from primary buffalo and sheep hepatocytes and compared with monolayer culture, cells cultured on poly-2-hydroxyethyl methacrylate (poly-HEMA)-coated non-adhesive plates, and the gold standard sandwich culture for hepatotoxicity study [51]. It was observed that spheroid cultures obtained from hanging drop methods were able to maintain nearly identical liver-specific functions when compared with the fresh primary hepatocytes, and they were thus established as the better model to produce spheroids from primary hepatocytes. Recently, in 2020, Cho et al. fabricated a pressure-assisted network for droplet accumulation (PANDA) system for cost-effective, consistent, fast, and mass production of spheroids by applying the hanging drop method [52]. A pressure drop was applied to the air chamber below the drop holding chamber, to assist the droplet to overcome the capillary force and form a stable drop without dripping to the pressure chamber. The proposed instrument effectively forms spheroids with minimal size variation thus could be a potential approach for fabrication of microtissues from liver cells for in vitro modeling application.

Thus, hepatic organoids can be recognized as the smallest three-dimensional unit that could closely represent the hepatic architecture. Though it possesses lower structural complexity compared to the organ, it can still find application in a wide range of disease modeling and screening of potential cures for human diseases.

### 3.4. Liver on a Chip

The goal of achieving in vivo mimicking in vitro models is brought closer to reality by the invention of microfluidic organs on a chip. The basic motivation for the fabrication of organs on a chip platform is to provide a dynamic microenvironment by applying physiologically relevant perfusion flow of nutrients to the cells to recapitulate the in vivo mimicking microenvironment and maintaining long-term organ-specific functionality for tissue engineering applications. Different designing software such as SOLIDWORKS, AutoCAD, etc. can be applied to design the chip and fabrication processes include Laser-Induced methods, 3D printing, photolithography, stereolithography, injection molding, etc. [53]. Bhise et al. reported the fabrication of a liver-on-a-chip platform by casting PDMS on predesigned poly (methyl methacrylate) (PMMA) molds [54]. Interestingly the fabricated device allows direct bioprinting of the liver cell spheroids made of HepG2/C3A cells by using negative PDMS wells mixed with GelMA hydrogel to the cell chamber. Different layers of the microfluidic chip were clamped together with screws and nuts which could be easily disassembled to gain direct access to the tissue construct. This feature is different from the conventional approach where different layers were permanently sealed, thus making access to the cultured tissue challenging. In the study, liver-specific biomarkers remained active throughout the 30 days of culture period and showed efficiency to the acetaminophen (APAP) toxicity analysis. Another group, Wang et al., described the application of microfluidic chip for in situ differentiation of iPSCs derived embryoid bodies (EBs) towards hepatic organoid culture and the organoid was eventually applied for dose and time-dependent toxicity of APAP. The device was fabricated by soft lithography of PDMS having a bottom layer with micropillar arrays and a top layer with inlet and outlet for introducing cells and culture media. A continuous flow of basic fibroblast growth factor (Bfgf, 10 ng Ml^−1^) and hepatocyte growth factor (HFG, 20 ng Ml^−1^) after formation of Ebs from day 5 to day 10 induced the effective formation of hepatic spheroids and were able to show all the liver-specific functions over 30 days. Although PDMS is the most reported material for fabrication of microfluidic chips due to its excellent gas permeability and optical clarity, it can absorb small hydrophobic molecules from the culture stream and these molecules eventually accumulate in bulk amounts inside the pores [55]. Thus, to overcome these challenges, many other materials have been considered for the construction of organs on a chip. Rennert et al. described the fabrication of an oxygen sensor integrated microfluidic chip made of cyclic olefin copolymers (COC)-TOPAS^®^ by injection molding process [56]. The liver organoids were prepared with HepaRG cells in combination with other essential non-parenchymal (LX-2 cells, Human umbilical vein endothelial cells, Peripheral Blood Mononuclear Cells, PBMCs cells) cells in a physiologically relevant ratio. The established liver model enables polarization which closely resembles the primary human liver tissue. Thus, microfluidic dynamic cultures in combination with liver organoids/spheroids can be fabricated as an effective in vivo mimicking in vitro model which can find further applicability in disease modeling, drug screening and many other tissue engineering approaches.

## 4. Application of Hepatic Organoids

### 4.1. Drug Screening

Drug screening for drug-induced liver injury by compounds is an essential part of drug development and is employed by pharmaceutical companies to evaluate the toxicological effect of drugs. Around one-third of the drugs fail during the screening process, which would eventually lead to a huge loss of opportunity to treat patients. Preclinical studies involve in vitro studies for drug efficacy, followed by in vivo and in vitro studies for its toxicological and metabolism study. To execute this, preclinical models have been developed by researchers to study drug-induced liver injury. Conventionally, in vitro preclinical studies are carried out on 2D cultured cellular systems in micro-well plates. However, 3D cell culture systems such as organoids have the potential to recapitulate the function and cellular composition of the organ better in comparison to 2D models. Moreover, hepatic organoids are stable for a longer period of time and are easy to carry out with live imaging. It also opens the possibility of patient-specific drug screening through the fabrication of organoids from the patients’ cells. Several hepatic organoids have been developed by researchers and have utilized platforms such as microfluidic systems for drug screening applications. In 2017, Broutier et al. accomplished that Catenin beta 1 mutant tumor-derived hepatic organoids demonstrate resistance to Wnt inhibitor LGK974 in contrast to bile duct-derived tumor organoids. The given studies were in accordance with the drug sensitivity of tumors present in patients [57].

Cytochrome P450 enzymes (CYP450s) are crucial for drug-based liver toxicity, and its induction in hepatic organoids to physiological level during differentiation makes organoids highly desirable for drug screening [58] and hepatic toxicological analysis. With the increase in demand for drug testing, high throughput methods for drug testing have been developed by the researchers to carry out multiple tests of drug screening in a single run. Microfluidic platforms have demonstrated their potential for organoids based high throughput drug screening. Sam et al. have reported the fabrication of hepatic organoids derived from HepG2 and NIH-3T3 fibroblasts. Further, the effect of acetaminophen on necrosis and apoptosis using a microfluidic platform was also studied [59].

In conclusion, hepatic organoids systems have proven their suitability for drug screening applications because of their liver function performance as well as sensitivity towards drugs/toxic substances. However, there are still challenges that need to be addressed to recapitulate the in vivo systems such as lack of control over morphology and composition of organoids as well as contrasting variation among batch to batch of organoids [60]. Variation among batches of organoids would limit the drug screening analysis reliability.

### 4.2. Disease Model

Animal-based disease models have been employed to explore the disease underlying mechanisms, drug efficacy, and hepatotoxicity. However, the use of animals as disease models brings ethical issues, requires highly skilled labor, and is time-consuming as well as expensive. Apart from this, animal models do not represent the native human systems and there is also significant variation among animals. To overcome this, in vitro models imitating the in vivo microenvironment have been developed. Initially, 2D cell culture systems were developed as liver disease models. However, as discussed earlier, hepatocytes have been observed to lose their phenotypic characteristics on 2D cultures making them significantly different from liver physiology. Over time, it was realized that 3D liver models are essential to simulate liver disease models which provide higher cell–cell interactions in comparison to 2D models and have the potential to maintain the liver phenotype as well as functional characteristics for a longer period, a requisite for a disease model. Liver disease progression involves interactions among multiple cell lineages such as HSCs, KCs, LSECs, and others because of which the interaction among cells of different lineages became an important aspect for the establishment of disease models. In a study by Liu and co-workers, hepatocytes and endothelial cells were cultured in different compartments of a microfluidic chip [61]. The given 3D culture system could have the potential to study the alcohol injury but lacks the cells responsible for the repair which involves HSCs and other immune cells. The presence of all the disease participating cells is critical to recreate the in vivo diseased conditions in vitro. In addition, the bioengineered construct should be able to maintain good viability, for which cell-biomaterial interactions have been explored by the researchers. Wang and his co-workers have fabricated a 3D liver construct from gelatin and hepatocytes [54]. In addition to this, the multicellular spheroid has been used to form a 3D liver disease model and has demonstrated hepatic stability for a longer period [62,63,64].

Disease models for alcoholic liver disease (ALD), non-alcoholic fatty liver disease (NAFLD) monogenic diseases, liver cancer, primary sclerosing cholangitis, viral hepatitis have been developed by researchers using single and multiple cell types. ALD is a chronic liver disease and caused by prolonged alcohol consumption. ALD development comprises three main characteristics, which involve fatty liver and steatohepatitis, followed by alcohol-based cirrhosis. Wang et al. have reported the fabrication of ALD model from ESC-derived expandable hepatic organoids co-cultured with human fetal liver mesenchymal cells and demonstrated variation in phenotype (ECM deposition, apoptosis, oxidative stress) as well as in gene expression (CYP2E1 and CYP3A4) connected to alcohol-related liver injury [65]. The given co-culture model can recapitulate a few aspects of disease progression. However, interaction and response of immune systems would have provided a better platform to imitate the ALD microenvironment. Lin et al. have integrated HepG2, LX-2, EAhy926, and U937 cells for the ALD model to mimic the liver sinusoid microenvironment. In addition to this, a significant variation in the expression of VE-cadherin, eNOS, VEGF, and α-SMA was observed when exposed to alcohol [66]. It is still challenging to imitate the various stages of ALD progression in vitro and requires an advanced, multimodal system with a controlled environment. Similar to ALD, NAFLD progression takes place through various stages involving steatosis, cirrhosis, hepatocarcinoma, fibrosis, and NASH. For NAFLD model, human pluripotent stem cells (hPSCs), human-induced hepatocytes (HiHeps), and hepatocyte stem cell lines (HepaRG) have been explored by researchers as potential cell lines for the fabrication of organoids. Yu et al. have fabricated HepaRG organoid to model NAFLD and shown lipid accumulation when exposed to free fatty acid (FFA) as well as change in glucose regulation and Akt phosphorylation. Further, the reduction in steatosis was observed when organoids were incubated with anti-steatosis compounds (pioglitazone hydrochloride and obeticholic acid) [67]. Sasaki and coworkers have also reported the development of mice primary hepatocytes derived from hepatic organoids for NAFLD. In addition, genetic markers for the progression of NAFLD to various stages of NASH in NAFLD models were identified [68]. For host-pathogen interaction studies, hepatic organoids infected with hepatitis B virus were studied to imitate the hepatitis B virus stages of the life cycle involving replication of HBV cDNA and its maintenance. In the given study, the response of non-parenchymal cells in addition to hepatocytes was recapitulated [69].

In totality, disease models from hepatic organoids have provided a better platform to integrate multiple cell types and perfusion systems to facilitate long-term stability. However, there are still challenges that need to be addressed such as liver zonation during and recapitulation of different stages of the disease.

### 4.3. Metabolism Prediction

The liver is the main organ for the regulation of carbohydrate, lipids, and amino acid metabolism, in addition to the synthesis of proteins. It is also central in regulating the conversion of ammonia to urea/bile, iron circulation levels, in addition to detoxification of antibiotics, alcohol, and drugs. Several everyday chemical entities have been developed by researchers with high hits and with structural analogy for drug development. Drug metabolism studies are essential for the pharmacokinetic studies of the new chemical entities [70]. Due to this, its understanding and the description of metabolism are required for the submission dossier in drug discovery programs to prevent drug attrition [71]. Metabolism-based in vitro studies were carried out by employing various platforms such as perfused liver, primary hepatocytes, liver slices, liver homogenates, subcellular liver fractions (S9, cytosol, microsomes), recombinant metabolizing enzyme (cytochrome P450 isozymes), and hepatic organoids [72,73,74]. However, hepatic organoids systems for the drug/xenobiotic/toxic substance metabolism studies are superior, owing to higher metabolism activity, high throughput analysis and cost effectiveness. Takanori et al. have investigated the hepatic clearance of drugs in liver spheroids derived from primary human hepatocytes. In the given study, PHH derived spheroids maintained their in vivo-like phenotype which bolstered the 7-days drug metabolism studies [75]. Apart from PHH, non-parenchymal cells (NPCs) also play a vital role in disease development and are required to mimic the in vivo conditions. Ouchi et al. has developed multicellular spheroids from iPSC derived hepatocytes, stellate and Kupffer-like cells to model inflammation and fibrosis [76]. In addition to this, Kumar et al. has established a disease model for non-alcoholic steatohepatitis (NASH), fibrosis, and cirrhosis involving iPSC derived hepatocytes and NPCs. TGF-β and fatty acid were employed to induce inflammatory reactions and fibrosis [44]. Microfluidic systems with perfusion systems would further enhance the metabolic activity of hepatic organoids. Active flow microenvironment has been observed to enhance the albumin synthesis as well as urea excretion in comparison to the static microenvironment. Such platforms have been used to analyze the Phase I and Phase II liver metabolism and study the first-pass drug metabolism through the integration of gut-like structure. To further enhance the sensitivity towards metabolism, Zhou et al. have fabricated microfluidic chambers coated with aptamer-coated electrodes to study the secretion of transcription growth factors [77]. Conclusively, hepatic organoid systems closely mimic the in vivo physiological microenvironment when integrated with active flow conditions along with the high metabolic activity.

### 4.4. Multi Organ-on-a-Chip

Platforms with multiple organs-on-a-chip have the potential to transform the conventional methods of research and healthcare development. Interaction between different organs is essential for the smooth functioning of the human body. The major communication between organs takes place through the blood and lymphatic system while the organs present are physically autonomous. The communication takes place through various means which involve soluble factors, exosomes, cells, hormones, etc. for homeostasis. Apart from this, entry of any foreign substance into our body involves its passage through various organs. For example, orally ingested substances are adsorbed through the intestine, followed by liver metabolization, delivered to the kidney via blood circulation. In addition, several diseases such as neurodegenerative diseases, osteoarthritis, fertility, gout, and sepsis are connected to multiple organs and require a multiple organ model system. Understanding communication among various organs would help us in the identification of biomarkers for diagnostic application. In particular, molecules such as miRNA, peptides, tumor-derived extracellular vesicles, tumor cells released by tumor tissue play an important role in metastasis.

Lee et al. have developed a gut-liver chip that imitates the absorption by the gut as well as metabolism by the liver. In the given study, fatty acid deposition was demonstrated in liver cells after absorption of fatty acid by the gut layer. In addition, the effect of tumor necrosis factor-α, butyrate, and α-lipoic acid on the fatty acid deposition/hepatic steatosis process was studied [78]. In another study by Sung and coworkers, the effect of turofexorate isopropyl and metformin on hepatic steatosis was studied on a gut-liver chip [79]. In another study, gastrointestinal tract and liver systems were developed for the toxicological analysis of polystyrene nanoparticles. The given platform had co-cultured enterocytes (Caco-2) and mucin-producing cells (TH29-MTX) which represent human intestinal epithelium and HepG2/C3A cells for the liver. Release of aspartate aminotransferase was observed after exposure to nanoparticles to demonstrate the liver injury by nanoparticles [80].

Fangchao et al. have reported a heart-liver chip derived from human induced pluripotent cells (hiPSC) and studied the effect of anti-depressant drug Clomipramine on the cardiac system as well as liver metabolism. Exposure to 1uM clomipramine with the liver chamber for 24 and 48 h has resulted in the loss of cell viability as well as compromised cardiac beating [81]. Apart from this, the lung-liver chip was also developed by other researchers. In a study by Julia and coworkers, MucilAir and HepaRG derived spheroids were co-cultured to evaluate the toxicity of inhaled substances. Exposure to the lung-liver chip to aflatoxin B_1_ has impaired the bronchial Mucilair tissues in monoculture. However, a protective effect was observed when co-cultured with hepatic spheroids, demonstrating the communication within the tissues [82]. Such models demonstrate multi-organs on a chip as a potential and more comprehensive tool for health research.

## 5. Conclusions

Liver is a very essential organ, with complex structural features and has multiple functions including metabolism, filtration, storage etc. Different liver diseases associated with environmental, dietary, or viral factors were identified over the years and the corresponding in vitro models were established for evaluation of disease mechanisms or finding effective drugs for treatment. Among these models, hepatic organoid-based liver tissue models with or without other tissues were established as a potent system for establishing in vivo mimicking microenvironment. Microfluidic platform and bioprinting of spheroids further improve the features of the in vitro model where the micro tissues were able to maintain all liver-specific functional properties such as albumin, urea secretion and cytochrome P450 activity efficiently over the required time frame. Many other features such as monitoring of oxygen gradient in the culture media by integrating oxygen sensors, or introduction of vascular network to mimic blood capillaries in hepatic organoids become other dimensions for further improvements. Thus, with fine tuning of liver microenvironments, the organoid culture models can act as a potential platform for in vitro modelling of human liver diseases and could find immense potential for drug and toxicology screenings to establish new therapeutic dimensions.

## Figures and Tables

**Figure 1 bioengineering-08-00185-f001:**
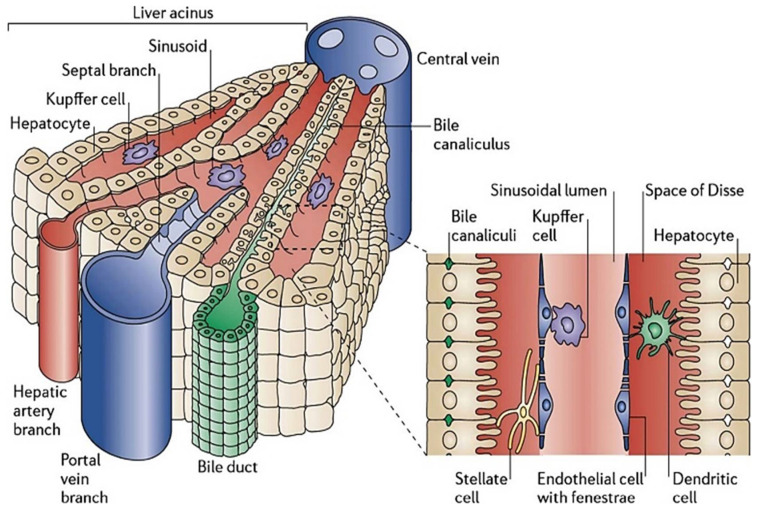
Structures of the liver lobule and liver sinusoids. Hepatocytes are aligned radially to form the liver plate along with the sinusoids. The portal veins and hepatic artery branches terminate in the sinusoids, draining blood into the sinusoids and through the acinus to the central vein. The sinusoids are lined by fenestrated liver sinusoidal endothelial cells with Kupffer cells interspersed onto the endothelium. Between the liver plate and the sinusoids is the space of Disse, containing extracellular matrix components and hepatic stellate cells [5].

**Figure 2 bioengineering-08-00185-f002:**
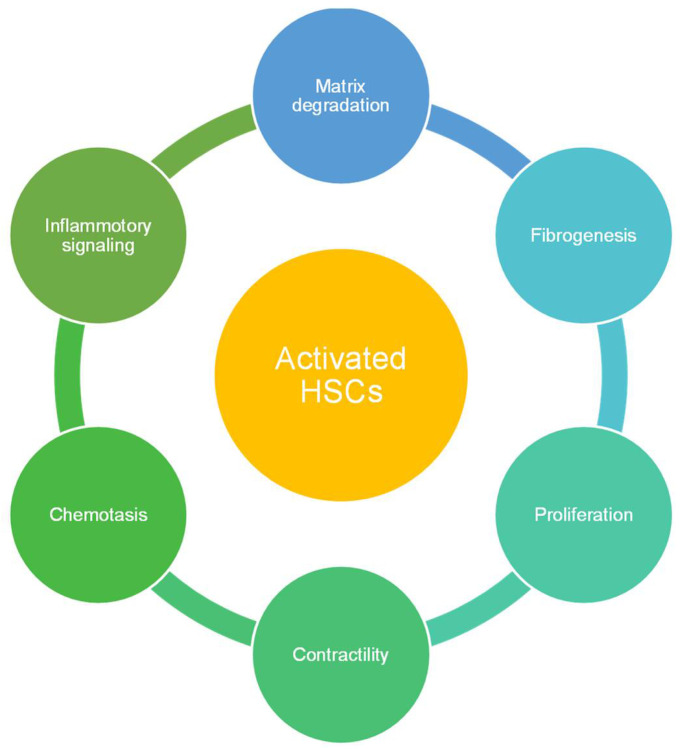
Functions of activated HSCs in diseased livers [19].

**Figure 3 bioengineering-08-00185-f003:**
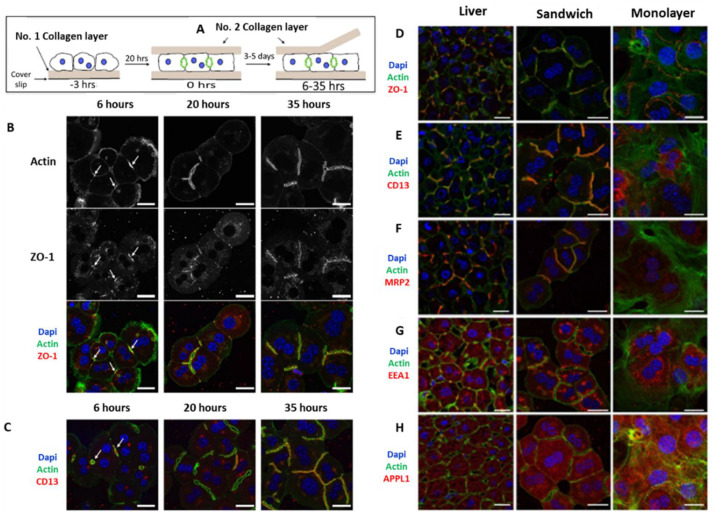
Cell polarity and bile canaliculi network formation in hepatocyte sandwich culture and the comparison of the endocytosis markers with liver slides, representing the in vivo conditions. (**A**) Protocol for fabrication of sandwich culture and removal of the soft collagen top layer for immunofluorescence staining. (**B**) Immunofluorescence images for re-establishment of cell polarity by staining markers for (**C**) tight junctions (ZO-1, F-Actin) and CD13, a transcytotic marker; A comparison of (**D**) tight junction markers (ZO-1, F-Actin) (**E**) cell polarity marker; (**F**) apical bile acid transporter (MRP2); and endosomal markers i.e., (**G**) EEA, Early endosome antigen 1, and (**H**) APPL1, adaptor protein containing a pleckstrin-homology domain, phosphotyrosine binding domain, and leucine zipper motif 1 between monolayer culture and sandwich culture was visualized and compared with liver slices. Scale bar = 20 µm [41].

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
