# Peer review of "3D Hepatic Organoid-Based Advancements in LIVER Tissue Engineering"

_bioengineering, 2021, doi:10.3390/bioengineering8110185_

Round 1

Reviewer 1 Report

The manuscript “Advances in hepatic organoid-based liver tissue engineering” by Panwar et al. is an important work that focuses on advances in hepatic organoid-based liver tissue engineering for applications mimicking liver diseases, screening drugs and developments in tissue engineering.

Although this manuscript raises very relevant points, it may be improved and, in my opinion, some additional publications could be described: - there is a lack related to monolayer co-cultures of  normal hepatocyte (rodent, human) and hepatocytes derived from induced pluripotent stem cells as well as publications using HepaRG cells; - why did the authors limit their analysis to sandwich cultures between layers of collagen and scaffold free spheroid/organoids and did not mention works of hepatic cells in scaffold-based systems (matrigel and other matrices…). Many publications, including their own work, have been done with various scaffolding matrices including collagen, gelatin, decellularize liver…; - in my opinion, the paragraph on bioprinting technology, liver on chip and/or microfluidic devices could be more detailed. The authors did a very good job using this technology with a gelatin-based 3D hydrogel and Huh7 cells.

Minor remark: Bhise et al. appears twice in the reference session: #54 and # 62

Author Response

Reviewer 01

The manuscript “Advances in hepatic organoid-based liver tissue engineering” by Panwar et al. is an important work that focuses on advances in hepatic organoid-based liver tissue engineering for applications mimicking liver diseases, screening drugs and developments in tissue engineering.

Although this manuscript raises very relevant points, it may be improved and, in my opinion, some additional publications could be described: - there is a lack related to monolayer co-cultures of  normal hepatocyte (rodent, human) and hepatocytes derived from induced pluripotent stem cells as well as publications using HepaRG cells; - why did the authors limit their analysis to sandwich cultures between layers of collagen and scaffold free spheroid/organoids and did not mention works of hepatic cells in scaffold-based systems (matrigel and other matrices…). Many publications, including their own work, have been done with various scaffolding matrices including collagen, gelatin, decellularize liver…; - in my opinion, the paragraph on bioprinting technology, liver on chip and/or microfluidic devices could be more detailed. The authors did a very good job using this technology with a gelatin-based 3D hydrogel and Huh7 cells.

Details added as per manuscript

 Minor remark: Bhise et al. appears twice in the reference session: #54 and # 62

Corrected

Reviewer 2 Report

Review article by Panwar et al. entitled “Advances in Hepatic organoid based liver tissue engineering” summarizes the importance of 3D hepatic organoid culture and its applicability the field liver toxicology and disease modeling. The review article is well written and text is clear and easy to read. The article provides a generalized background of the topic that gives the reader an appreciation of the wide range of applications but I have major reservations before considering for the publications.

General comments:

  1. Overall article provided the overview of the current scenario but multiple important recent important studies are not included.
  2. Currently iPSCs based organoids culture are gaining importance only few studies are included the article. iPSCs based culture forms a major chunk in advances in organoid culture should be included in the review.
  3. The word ‘regenerative medicine’ should be removed from the abstract as review does not address anything related to regenerative medicine.
  4. Section 2: subsection 2.1 ‘Functional units of the liver’ can be reduced.
  5. May be restructure the title (if possible)

Specific comments:

  1. There are few mistakes which include typos in the text, also in many place short forms are used but never mentioned earlier e.g. page 2; HEPs or HCs.
  2. Also on page 3 line 115-116 ; authors mentioned Embryonic stem cells(ESCs) or Pluripotent stem cells(PSCs); ESCs are also PSCs; does authors wants to say iPSCs: it makes a lot of difference.
  3. On the same page authors mentioned that HSCs constitutes 15% of the total cell present in liver, which is not true; would recommend it review again.
  4. Page 4 line 150; ‘as shown in hsc1’ please correct.
  5. Liver Sinusoidal Endothelial cells sections should be reviewed thoroughly and provide latest new insight e.g. SE-1 and CD31 were markers identified for LSECs but in recent publications described newer and better marker for identifying these cell types.
  6. Line 198 typo
  7. Line 277; authors mentioned CD36 for Kupffer cells but CD36 is not a Kupffer cell marker should be corrected.
  8. Line 281; reference Hendriks et al. is missing
  9. In the section 3 : Development of hepatic tissue : Strategies and challenges. Authors have not included any studies related to many of the organoids culture developed recently (PHH based organoids/ HepaRG and stellate cells) and almost no study was mentioned about iPSCs based organoid culture. Therefore recommend authors to review and include recent and important papers iPSCs by Takebe et al. Nature, 2013; Kumar et al. Biomaterial 2021.
  10. Section 4: Authors have not performed extensive literature search excluded the studies published form and not included recent studies with iPSCs (Like Ouchi et al. Nature metabolism 2020; Kumar et al. Biomaterials 2021).
  11. In the subsection 4.3; Authors should also include the metabolization studies not only with microfluidics system but also included other organoid/ spheroid models as well.
  12. In multiple places, last author of the article is addressed first e.g. ‘last authors and colleagues’ should be addressed in rightly while referring in the text.

Author Response

Reviewer 02

Review article by Panwar et al. entitled “Advances in Hepatic organoid based liver tissue engineering” summarizes the importance of 3D hepatic organoid culture and its applicability the field liver toxicology and disease modeling. The review article is well written and text is clear and easy to read. The article provides a generalized background of the topic that gives the reader an appreciation of the wide range of applications but I have major reservations before considering for the publications.

General comments:

  1. Overall article provided the overview of the current scenario but multiple important recent important studies are not included.

Updated with recent studies

  1. Currently iPSCs based organoids culture are gaining importance only few studies are included the article. iPSCs based culture forms a major chunk in advances in organoid culture should be included in the review.

iPSC  studies have been added to the section

  1. The word ‘regenerative medicine’ should be removed from the abstract as review does not address anything related to regenerative medicine.

Removed

  1. Section 2: subsection 2.1 ‘Functional units of the liver’ can be reduced.

Changed to - Liver physiological Units

  1. May be restructure the title (if possible)

Restructured

“3D Hepatic organoid-based advancements in Liver tissue engi-neering”

Specific comments:

  1. There are few mistakes which include typos in the text, also in many place short forms are used but never mentioned earlier e.g. page 2; HEPs or HCs.

Corrected

  1. Also on page 3 line 115-116 ; authors mentioned Embryonic stem cells(ESCs) or Pluripotent stem cells(PSCs); ESCs are also PSCs; does authors wants to say iPSCs: it makes a lot of difference.

Corrected to ESC only

  1. On the same page authors mentioned that HSCs constitutes 15% of the total cell present in liver, which is not true; would recommend it review again.

Corrected

  1. Page 4 line 150; ‘as shown in hsc1’ please correct.

Corrected

  1. Liver Sinusoidal Endothelial cells sections should be reviewed thoroughly and provide latest new insight e.g. SE-1 and CD31 were markers identified for LSECs but in recent publications described newer and better marker for identifying these cell types.

Corrected with of LYVE1, STAB2, PECAM1 and CD32B markers

  1. Line 198 typo

Corrected

  1. Line 277; authors mentioned CD36 for Kupffer cells but CD36 is not a Kupffer cell marker should be corrected.

Corrected

  1. Line 281; reference Hendriks et al. is missing

Corrected

  1. In the section 3 : Development of hepatic tissue : Strategies and challenges. Authors have not included any studies related to many of the organoids culture developed recently (PHH based organoids/ HepaRG and stellate cells) and almost no study was mentioned about iPSCs based organoid culture. Therefore recommend authors to review and include recent and important papers iPSCs by Takebe et al. Nature, 2013; Kumar et al. Biomaterial 2021.

Added

  1. Section 4: Authors have not performed extensive literature search excluded the studies published form and not included recent studies with iPSCs (Like Ouchi et al. Nature metabolism 2020; Kumar et al. Biomaterials 2021).

Added

“Apart from PHH, non-parenchymal cells (NPCs) also play a vital role in disease de-velopment and are required to mimic the in vivo conditions. Ouchi et. al. has developed multicellular spheroids from iPSC derived hepatocytes, stellate and Kupffer like cells to model inflammation and fibrosis [77]. In addition to this, Kumar et. al. has established a disease model for non-alcoholic steatohepatitis (NASH), fibrosis, and cirrhosis involv-ing iPSC derived hepatocytes and NPCs. TGF-β and fatty acid were employed to induce inflammatory reactions and fibrosis [45].”

  1. In the subsection 4.3; Authors should also include the metabolization studies not only with microfluidics system but also included other organoid/ spheroid models as well.

Spheroids system for drug metabolism studies have been added

  1. In multiple places, last author of the article is addressed first e.g. ‘last authors and colleagues’ should be addressed in rightly while referring in the text.

Corrected

Round 2

Reviewer 2 Report

A minor correction in Line 287 the reference should be correctly written it should be Kumar et al instead of Manoj et al